# Simulations of the Ultra-Fast Kinetics in Ni-Si-C Ternary Systems under Laser Irradiation

**DOI:** 10.3390/ma14164769

**Published:** 2021-08-23

**Authors:** Salvatore Sanzaro, Corrado Bongiorno, Paolo Badalà, Anna Bassi, Ioannis Deretzis, Marius Enachescu, Giovanni Franco, Giuseppe Fisicaro, Patrizia Vasquez, Alessandra Alberti, Antonino La Magna

**Affiliations:** 1CNR-IMM, Zona Industriale Strada VIII 5, 95121 Catania, Italy; salvatore.sanzaro@imm.cnr.it (S.S.); corrado.bongiorno@imm.cnr.it (C.B.); ioannis.deretzis@imm.cnr.it (I.D.); giuseppe.fisicaro@imm.cnr.it (G.F.); alessandra.alberti@imm.cnr.it (A.A.); 2STMicroelectronics, Zona Industriale Stradale Primosole 50, 95121 Catania, Italy; paolo.badala@st.com (P.B.); anna.bassi@st.com (A.B.); giovanni.franco@st.com (G.F.); patrizia.vasquez@st.com (P.V.); 3Center for Surface Science and Nanotechnology, University Politehnica of Bucharest, Splaiul Independentei nr. 313, AN031, District 6, 060042 Bucharest, Romania; marius.enachescu@cssnt-upb.ro

**Keywords:** Ni-Si-C ternary system, laser annealing, finite element method

## Abstract

We present a method for the simulation of the kinetic evolution in the sub µs timescale for composite materials containing regions occupied by alloys, compounds, and mixtures belonging to the Ni-Si-C ternary system. Pulsed laser irradiation (pulses of the order of 100 ns) promotes this evolution. The simulation approach is formulated in the framework of the phase-field theory and it consists of a system of coupled non-linear partial differential equations (PDEs), which considers as variables the following fields: the laser electro-magnetic field, the temperature, the phase-field and the material (Ni, Si, C, C clusters and Ni-silicides) densities. The model integrates a large set of materials and reaction parameters which could also self-consistently depend on the model variables. A parameter calibration is also proposed, specifically suited for the wavelength of a widely used class of excimer lasers (*λ* = 308 nm). The model is implemented on a proprietary laser annealing technology computer-aided design (TCAD) tool based on the finite element method (FEM). This integration allows, in principle, numerical solutions in systems of any dimension. Here we discuss the complex simulation trend in the one-dimensional case, considering as a starting state, thin films on 4H-SiC substrates, i.e., a configuration reproducing a technologically relevant case study. Simulations as a function of the laser energy density show an articulated scenario, also induced by the variables’ dependency of the materials’ parameters, for the non-melting, partial-melting and full-melting process conditions. The simulation results are validated by post-process experimental analyses of the microstructure and composition of the irradiated samples.

## 1. Introduction

Ultra-fast heating/cooling cycles (in the sub-10^−6^ s time scale) induced by pulsed laser irradiation with pulses of electro-magnetic power density in the ten-hundred ns range is, nowadays, relevant technology for the production of advanced materials, manufacturing of electronic devices and other high-tech applications [1,2]. The benefits of this annealing technique are localized-in-space heat sources (due to the nm range of the laser penetration depth in key-materials), very high peak temperatures in the source location, fast quenching with a negligible thermal budget in material regions tens of microns from the heat source [3]. The extreme non-equilibrium conditions induced by irradiation also allow for obtaining materials/compounds which are otherwise hard or even impossible to form, such as magnetic alloys with customized magnetic properties [4], including quenched nematic order [5].

In the microelectronics field, thanks to these characteristics, Laser Annealing (LA) is, e.g., the privileged doping solution in process optimization studies for vertically integrated device structures; whereas, the ultra-fast transient melting condition is used for both healing the crystal damage and activating the impurity profile [2]. Melting with melt depths/areas/volumes in the nanoscale range is a key feature of LA activating phenomena, which are too slow or simply not occurring in the solid phase. As a consequence, the usual assumption is that in pulsed laser annealing, process mass transport does not occur in the non-melting regime, and therefore, LA modeling focuses on the prevalence of melting conditions.

Actually, the Ni-Si system, of crucial interest for the metal contact microelectronic technology [6], is a well-known exception to this assumption. Indeed, a fast inter-diffusion of the two elements can be activated by a low thermal budget, which favors the formation of Ni-silicide compounds in the solid phase at the time scale of tens of ns [7,8,9]. The complex interplay between solid-phase and liquid-phase kinetics makes the accurate simulation of the LA process in the Ni-Si system challenging, and few examples are reported in the literature, where only partial models are applied (see Ref. [8] and references therein).

The increasing importance of silicon carbide (SiC) as an innovative semiconductor material for device applications has recently shifted the focus from Ni-Si to the Ni-Si-C ternary system. Indeed, also for SiC devices, Ni-silicides are key contact materials, and LA could be applied, again thanks to the localized heating features, as a thermal process for the contact formation [10,11]. Of course, the still immature simulation techniques, tentatively introduced in the case of Ni-Si systems, need to be completely readdressed when dealing with the simulation of the silicide formation kinetics on a SiC semiconductor substrate.

In this paper, we propose a continuum simulation model specifically developed for the laser annealing simulation of structures based on the Ni-Si-C ternary system. The model potentially considers the whole thermodynamic and kinetic conditions for these materials and is able to deal with the complex phase diagram of the Ni-Si-C system, which is characterized by alloys and compound phases. The model is implemented in a proprietary simulation tool for the numerical solutions, and in the following, we will discuss examples for an experimentally relevant case. The richness of the kinetic scenario shown by the results, as a function of the irradiation parameters, is somehow surprising, also indicating the route for possible technological solutions for the applications.

## 2. Materials and Methods

Predicting the heating, eventual melting and structural phase transitions of a Ni-Si-C system under irradiation is the goal of the simulation method herein. The reference laser process is pulsed (single pulse or multiple pulses and space uniform beam with fixed wavelength) with a pulse duration of 100–200 ns and a repetition rate of the order of 1–10 Hz. With no lack of generalization, we consider the wavelength of the XeCl laser (*λ* = 308 nm); only the calibration’s modifications are necessary for the application of the approach to another laser type.

The evolution model for the Ni-Si-C system is formulated in the framework of the phase-field model for the phase transition with temperature change driven by irradiation in the thermalization approximation. The reference model (e.g., [12] and the references therein) is based on self-consistent solutions of (a) the time-harmonic Maxwell equation for the calculation of the local heat source where the incident wave is characterized by the wavenumber $k_{0}$, the laser fluence $E_{\mathrm{dens}}$ and the time dependence of the power pulse, (b) the Fourier law for the simulation of the temperature field evolution and (c) the phase-field equation simulating the evolution of the solid-liquid front in the case of a melting process. The model is mathematically formulated in terms of coupled non-linear partial differential equations (PDE), which can be solved numerically with a Finite Element Method (FEM) in a proper simulation mesh [13]. The self-consistency is related to the cross dependence between the fields (i.e., the variables of the PDEs) and the physical parameters ruling their behavior (e.g., the optical constants in the Maxwell equations can depend on temperature and phase).

Starting from the reference phase-field model, we have formulated and implemented a model specifically suited for the Ni-Si-C system. The challenge is related to the complexity of this ternary system (e.g., [14]) characterized by the stability of different compounds (especially the silicide class: Ni_x_Si_y_) and the non-trivial role of the carbon present in the intermixed stage. Moreover, as opposed to other systems (e.g., Si-Ge), the kinetic evolution of the Ni-Si-C system is also fast in the solid phase, due to the possibility of a strong intermixing at high temperature in the ns regime and relative formation of Ni_x_Si_y_ compounds due to solid-state reactions [8]. This solid-state evolution, occurring during the early stage of the LA process, non-trivially affects the eventual subsequent melting stage due to the strong dependence of the melting point on the local composition (elements + compounds).

The model has been integrated into an open-source PDE solver [13], it is based on 13 evolving fields (and relative PDEs, see Equations 1–13 below): Electron magnetic field E (no ferromagnetic materials and transverse electric polarization considered in the present study), Temperature T, Phase φ, Ni $C_{Ni}$, Si $C_{Si}$, C $C_{C}$, C clusters $C_{C-clust}$, 6 classes of Ni-Si compounds $C_{Ni_{x}Si_{y}}$ with $Ni_{x}Si_{y}$ = $\left\{\;NiSi_{2},NiSi,\;Ni_{3}Si_{2},Ni_{2}Si,\;Ni_{5}Si_{2},Ni_{3}Si\right\}$. We assume that the silicide reactions chain is activated by the $Ni_{3}Si$ nucleation with a rate proportional to the local density of the minority element (Si or Ni), then the reactions’ chain proceeds according to the subsequent Si enrichment with a single reaction branching towards the $Ni_{5}Si_{2}$ variable ($Ni_{5}Si_{2}$ class in the following), as other compound classes effectively regroup different silicide structures with similar stoichiometry rates (close to 2.5 in this case, e.g., as the $Ni_{31}Si_{12}$ compound). The model equations are:(1)$$\nabla \times \left(\nabla \times E\right)-k_{0}^{2}\left(\varepsilon_{r}-j\varepsilon_{i}\right)E=0$$
(2)$$\rho c_{p}\frac{\partial T}{\partial t}=\nabla \left[k\left(T,\varphi \right)\nabla T\right]+30\varphi^{2}\left(1-\varphi^{2}\right)L\frac{\partial \varphi }{\partial t}+Q$$
(3)$$\tau_{\varphi }\times \frac{d\varphi }{dt}=W^{2}\nabla^{2}\left(\varphi \right)-2\varphi \left(1-\varphi \right)\left(1-2\varphi \right)-8\;\lambda \left(T,\varphi \right)\left[T-T_{m}\right]\varphi^{2}{\left(1-\varphi \right)}^{2}$$
(4)$$\frac{\partial C_{Ni}}{\partial t}=\nabla \left(D_{Ni}\nabla C_{Ni}\right)-3\times \theta_{Ni}\frac{C_{Ni}}{\tau_{Ni_{3}Si}}-3\times \theta_{Si}\frac{C_{Si}}{\tau_{Ni_{3}Si}}$$
(5)$$\frac{\partial C_{Si}}{\partial t}=\nabla \left(D_{Si}\nabla C_{Si}\right)-\theta_{Ni}\frac{C_{Ni}}{\tau_{Ni_{3}Si}}-\theta_{Si}\frac{C_{Si}}{\tau_{Si_{tot}}}$$
(6)$$\frac{\partial C_{C}}{\partial t}=\nabla \left(D_{C}\nabla C_{C}\right)-\theta_{C}\frac{C_{C}}{\tau_{C}}$$
(7)$$\frac{\partial C_{Ni_{3}Si}}{\partial t}=\theta_{Ni}\frac{C_{Ni}}{\tau_{Ni_{3}Si}}+\theta_{Si}\frac{C_{Si}}{\tau_{Ni_{3}Si}}-5\times \theta_{Si}\frac{C_{Si}}{\tau_{Ni_{5}Si_{2}}}-2\times \theta_{Si}\frac{C_{Si}}{\tau_{Ni_{2}Si}}$$
(8)$$\frac{\partial C_{Ni_{5}Si_{2}}}{\partial t}=3\times \theta_{Si}\frac{C_{Si}}{\tau_{Ni_{5}Si_{2}}}$$
(9)$$\frac{\partial C_{Ni_{2}Si}}{\partial t}=3\times \theta_{Si}\frac{C_{Si}}{\tau_{Ni_{2}Si}}-3\times \theta_{Si}\frac{C_{Si}}{\tau_{Ni_{3}Si_{2}}}$$
(10)$$\frac{\partial C_{Ni_{3}Si_{2}}}{\partial t}=2\times \theta_{Si}\frac{C_{Si}}{\tau_{Ni_{3}Si_{2}}}-\theta_{Si}\frac{C_{Si}}{\tau_{NiSi}}$$
(11)$$\frac{\partial C_{NiSi}}{\partial t}=2\times \theta_{Si}\frac{C_{Si}}{\tau_{NiSi}}-\theta_{Si}\frac{C_{Si}}{\tau_{Ni_{2}Si}}$$
(12)$$\frac{\partial C_{Ni_{2}Si}}{\partial t}=\theta_{Si}\frac{C_{Si}}{\tau_{Ni_{2}Si}}$$
(13)$$\frac{\partial C_{C-clust}}{\partial t}=\theta_{C}\frac{C_{Si}}{\tau_{C}}$$
where:(14)$$\frac{1}{\tau_{Si_{tot}}}=\frac{1}{\tau_{Ni_{3}Si}}+\frac{1}{\tau_{Ni_{5}Si_{2}}}+\frac{1}{\tau_{Ni_{2}Si}}+\frac{1}{\tau_{Ni_{3}Si_{2}}}+\frac{1}{\tau_{NiSi}}+\frac{1}{\tau_{Ni_{2}Si}}$$
(15)$$\theta_{Ni}=\left\{\begin{array}{c} 1\;if\;C_{Ni}<C_{Si}\\ 0\;otherwise \end{array}\right.$$
(16)$$\theta_{Si}=\left\{\begin{array}{c} 1\;if\;C_{Ni}\geq C_{Si}\;and\;C_{Ni}<3\times C_{Si}\\ 0\;otherwise \end{array}\right.$$
(17)$$\theta_{C}=\left\{\begin{array}{c} 1\;if\;C_{C}>C_{Si}\;\\ 0\;otherwise \end{array}\right.$$

We use the following values of the phase-field parameters $\tau_{\varphi }=4.83\times 10^{-9}\;s$ and W=1 nm while $\lambda \left(T,\varphi \right)$ is a phase-field function determined according to the Karma-Rappel prescription: i.e., imposing that the diffuse interface model reproduces the sharp interface limit and the latent heat balance at the moving interface correctly [12]. We note that the factors of the non-linear phase-dependent term in Equations (2) and (3) are related to the particular phase field formulation applied (see [12] and the references therein for the derivation) with a phase function that recovers the liquid (solid) properties at the φ=0 (φ=1) value. The speed law, as a function of the under/over cooling, is implemented in the $\lambda \left(T,\varphi \right)$ expression is Fulcher-Vogel type [15]:(18)$$V\left(T\right)=\exp\left(-\frac{E_{a}}{kT}\right)\times \exp\left\{1-exp\left[\left(\frac{\rho L}{k_{b}N}\right)\left(\frac{1}{T_{M}}-\frac{1}{T}\right)\right]\right\}$$

All the model parameters, which in general could depend on temperature, atomic fraction, and phase, are described in Table 1, Table 2, Table 3 and Table 4. In these tables, the parameters’ values and/or their dependencies from the model variables are also reported. A key feature of the model approach is the dependence of the local values of the physical properties on the local composition. We have used a single interpolation variable *X* to rule this dependence. In the results, which will be discussed in the following, we have made this variable equivalent to the local Ni atomic fraction: i.e., for *X* = 1, the properties will be equal to the pure Ni; while for *X* = 0, they will be SiC. We note then this is not a mandatory condition, and a different functional choice on other model variables could be implemented in a future refined calibration of the model. In agreement with the ternary Ni-Si-C thermodynamic phase diagram [14], in the liquid phase, only the carbon atoms forming the carbon cluster are not affected by melting: i.e., we assume the full dissolution of silicide compounds and complete miscibility of the residual C atoms (i.e., the non-clustered fraction) and of the Si and Ni atoms present in the solid elements and compounds. We note that the same functional dependence is assumed for the alloy diffusivity coefficient ruling the intermixing for the three elements, which vary as a function of the phase φ (solid-liquid) and the alloy fraction parameter *X* (see Table 4).

## 3. Results

We have integrated the continuum model presented in the previous section in a custom TCAD tool based on an open-source FEM framework and mesh generator dedicated to laser processing [13]. The related numerical simulation reveals complex features, which critically depend on the process parameters [26] (fluence or power density and laser pulse duration and shape) and the initial system preparation (i.e., the components’ space distribution in the initial state). In this paper, we will discuss 1D systems; anyhow, the code works starting with 2D and 3D material distribution. In order to study a case of relevance for the experimental application, we considered an “as prepared” system formed by a stack of air, thin pure Ni film and a 4H-SiC substrate. The experimental counterpart is the deposited Ni layer on 4H-SiC which undergoes the thermal processes for the formation of back-contacts in 4H-SiC based devices [10]. Therefore, the element fraction variables in the initial states of the model are $C_{Ni}=1$ in the Ni layer, and $C_{Ni}=0$ elsewhere, $C_{Si}=0.5$ in the SiC substrate and $C_{Si}=0,\;\;C_{C}=0.5$ in the SiC substrate and $C_{C}=0$ elsewhere. The other variables of the initial simulation state are all constant in the simulation box: uniform temperature field at RT T≡300 K, solid phase-field φ≡1, null clustered carbon component $C_{C-clust}\equiv 0$, null silicide components $Ni_{x}Si_{y}\equiv 0$. A zero flux (Neumann) boundary condition is imposed on the top (air) boundary. The incident laser wave is coming from the top of the air in a descending direction.

In order to qualitatively categorize the simulation scenario, we can distinguish three main cases which subsequently occur when increasing the laser power density: non-melting, partial melting, and full-melting regimes. In the following subsections, we will discuss the simulation features in these three cases.

### 3.1. Sub-Melting Regime

If the laser fluence $E_{dens}$ is below a (case dependent) threshold, no liquid-solid transient phase transition occurs, and all the kinetic evolution is generated by solid-phase element intermixing and reactions. Of course, the evolution of the fast-varying temperature field is the key aspect driving all the material kinetics. In Figure 1, we show the maximum temperature $T_{max}$ (obtained for this case at the surface location) and the power absorbed by the laser pulse as a function of the time in a system with an initial 100 nm thick Ni film and 110 µm thick 4H-SiC substrate for an irradiation process with an energy density of 2.2 J/cm^2^, which is below the melting threshold. We notice that, for consistency, we will consider the same power density pulse in the simulations discussed in this paper: the change of the pulse shape and, especially, its duration quantitatively affects the results, but the overall scenario is usually confirmed.

Comparing the absorbed electro-magnetic power and $T_{max}\left(t\right)$, we can observe the typical shift of the two temporal profiles related to the heat diffusion in the sample, e.g., in the interval between ≈110 and ≈190 ns, despite the reduction of the laser power density, the temperature continues to rise. From ≈190 to ≈310 ns, the system begins the thermal quenching since the laser is now too weak to sustain the local temperature increase against the heat diffusion. After ≈310 ns, the laser’s power reaches the zero value, and the temperature continues to decrease, again reaching RT = 300 K after a few tens of microseconds (not shown).

Intermixing between Ni, Si and C atomic species is the first thermally activated phenomenon characterizing the early state of the annealing. In Figure 2a, a snapshot of the elements’ volume fraction after 100 ns of simulated evolution for the $E_{dens}$ = 2.2 J/cm^2^ process is shown. Calibrated simulations (see Table 4 for the intermixing coefficient) predict a relevant and asymmetric (stronger in the Ni-rich region in comparison with the SiC one) intermixing of the atomic species in the pure solid phase during the early stage of the heating process, with Si and C diffuse profiles in the Ni-rich regions.

A relatively high solid solubility level of Si (close to 20%) can be obtained at high temperatures (the maximum temperature is $T_{max}\left(t=100\;\mathrm{ns}\right)\sim 1000\;K$ for the case in the snapshot in Figure 2a while the maximum temperature obtained is more than 1400 K a discussed in the following) in the Ni crystalline matrix whereas C monomers tend to segregate in C aggregates according to an effective rate (low in the pure solid phase) ruled by the reaction and diffusion coefficients. A tail of the Ni profile progressively extends in the SiC-rich region, increasing the reactivity of the interfacial layer, where the three atomic species are concurrently present in similar proportions. As a consequence of this penetration, the Ni-rich region at the end of the process is thicker than the original one, e.g., ≈120 nm in Figure 2b for a $E_{dens}=2.2\;{J/\mathrm{cm}}^{2}$ process, to be compared with the 100 nm thickness of the deposited Ni layer.

In the Ni-SiC interfacial region, the unbalance of thermodynamic stability between solid-phase solution state and the $Ni_{x}Si_{y}$ compounds’ state leads to the formation of silicide compounds; of course, if the relative local stoichiometry of the Ni and Si components and the local temperature permit these transformations with a significant rate in relation to the time scale of the process. These conditions are ruled both by the model Equations (4)–(17) and the calibrated reaction coefficient expressions (Table 4). In Figure 3, the local density profiles of silicide compounds after the process obtained by means of the laser annealing simulations of the processes at 2.2 J/cm^2^ are shown together with the maximum value of the local temperature field $T^{max}\left(x\right)$ as a function of the position obtained during the simulated annealing. We show only the Ni_3_Si, Ni_5_Si_2_ and Ni_2_Si since the density of other compounds is less significant (less than 0.01 in the used scale). We note that, in this case, and whenever we plot compound density profiles in this paper, the local fraction of the Ni and Si elements in the compounds can be obtained from the plotted quantity by simply multiplying this quantity by the respective stoichiometry coefficients.

The silicide with the higher Ni content (Ni_3_Si) can be considered as the “nucleation” phase, and it is the only one that really characterizes the silicide phases in the sub-melting regime. The density peak is generated by the term in Equation (7) multiplied by $\theta_{Ni}$. Therefore, the only one active when Ni is a minority element with respect to Si (i.e., in the SiC region, see also Equation (15)). We note that in the early calibration proposed here, we do not distinguish the reaction rate for the first two terms in the right side of Equation (15), i.e., we use the same $\tau_{Ni_{3}Si}$ expression for the Ni_3_Si nucleation also when Si is the minority element. Further focused investigations, with comparisons between simulations and experimental data, are necessary in order to, eventually, revise this choice.

The local temperature in the silicide region exceeds 1400 K (Figure 3 right axis) during the process; this is a sufficient thermal budget to activate the silicide formation. Other silicide compounds form in the (expanding) Ni-rich region (i.e., when Si is the relative minority element) at significantly lower rates, and they appear in the simulations at negligible densities (see Ni_5_Si_2_ and Ni_2_Si profiles in Figure 3).

### 3.2. Partial-Melting Regime

The transient melting phenomenon is simulated for fluences $E_{dens}\geq E_{treshold}≅2.3\;{J/\mathrm{cm}}^{2}$. The melting partially (partial-melting regime) affects the Ni-rich layer in the energy density range $2.3\;{J/\mathrm{cm}}^{2}\leq E_{dens}<3.0\;{J/\mathrm{cm}}^{2}$. In Figure 4, a snapshot of the phase, temperature and total Ni atomic fraction taken during the melting stage, i.e., after 220 ns of the simulated irradiation process, is shown for a process with $E_{dens}=2.5\;{J/\mathrm{cm}}^{2}$. In this case, the melting starts at t_on-melt_ ≈ 120 ns and ends at t_off-melt_ ≈ 240 ns (the time origin is the irradiation pulse starting, see Figure 1) and the maximum molten extension, calculated by the φ<0.5 condition, is ≈35 nm.

It is clear from Figure 4 that the liquid phase nucleates at the Ni-SiC interface for this value of the fluence in spite of the larger value of the temperature obtained at the surface (*x* = 0). This unusual inner location of the melting is caused by the solid phase intermixing, occurring during the heating in the *t* < *t_on-melt_* interval of the irradiation, which tends to lower the melting point of the ternary system with respect to the pure Ni phase. Indeed, in the binary phase diagram of the Ni-Si system, for pure Ni, the melting temperature is 1728 K while it reaches the value of 1424 K for a Si:Ni stoichiometric ratio of about 0.2, and a similar trend is observed in the Ni-Si-C system [14]. We have considered this effect in the implemented calibration dependence of $T_{M}\left(X\right)$ and the simulated peculiar behavior. It is a direct kinetic consequence when the temperature difference (T = 1523 K at the surface and T = 1493 K at the Ni-SiC interface at *t* = *t_on-melt_* for the $E_{dens}=2.5\;{J/\mathrm{cm}}^{2}$ process) between the surface and the Ni-SiC interface location (where the intermixing is maximal), is not large enough to compensate the $T_{M}\left(X\right)$ reduction. Melting occurs at this interface position.

The diffusivity coefficient increases from maximum values in the order of 10^−10^ m^2^/s in the solid phase to ≈2 × 10^−8^ m^2^/s values in the liquid phase (see Table 4). This jump of the intermixing coefficient significantly alters the element profiles in the molten regions (Ni profile, Figure 4). Moreover, as stated in Section 2, the silicide compounds, which form in the *t* < *t_on-melt_* pre-melting stage, are dissolved by the mobile melting front; e.g., the Ni profile portion in the molten region of Figure 4 is fully composed of Ni monomers in the Ni-Si liquid alloy coexisting with the C (monomers and clusters) counterpart.

The energy density range where the transient melting occurs at the Ni-SiC interface is $2.3{\;J/\mathrm{cm}}^{2}\leq E_{dens}\leq 2.8\;{J/\mathrm{cm}}^{2}$. We note that, in these conditions, the melting event cannot be easily detected with in-situ reflectivity measurements since the phase transition could be too deep to be probed with surface reflectivity.

Ni silicide phases reform after the transient melting, during the quenching stage of the process. However, the strong element intermixing obtained in the liquid phase brings, in the few tens of ns of the transient melting duration, the local composition to a completely different state. As a consequence, the solid-state reactions activated after the melting can follow a different path with respect to the non-melting regime. In particular, simulations show the increased relevance of compounds with higher Si content. In Figure 5, we show some snapshots of the simulated local density profiles of three silicide compounds (profiles of compounds with higher Si content with respect to the Ni_2_Si have a maximum lower than 0.01 in this scale in the whole range of conditions explored in this paper) during the melting and regrowth stages for the laser annealing process at the 2.6 J/cm^2^ energy density; whilst the final profiles after the full quenching at the end of the process are shown in Figure 6. The phase function is also plotted in the two figures. Similar to the non-melting case reported in Figure 3, we observe a sharp peak of Ni_3_Si density at the interface of the SiC material. The presence of the Ni_3_Si phases in this particular position have already been discussed for the non-melting case. This Ni-rich silicide is again the most important one after the partial melting process. However, we can observe that the liquid phase intermixing also activates a relevant formation of silicide compounds with higher Si content, which distribute preferentially in different positions along the depth.

For fluence larger than $2.8\;{J/\mathrm{cm}}^{2}$ the liquid phase more conventionally nucleates at the surface. Starting from this value of the fluence, the melting extension (melt depth in this case) increases strongly with the fluence, and the full melting of the Ni-rich layer occurs at $3.0\;{J/\mathrm{cm}}^{2}$. The surface melting can give access to the in-situ reflectivity measurements studied in this regime of fluence.

### 3.3. Full-Melting Regime

In Figure 7, the melting extension $D_{max}\left(E_{dens}\right)$ as a function of the fluence is shown in the 2.0–3.8 J/cm^2^ energy density range. The different regimes (non-melting, partial melting at the Ni-SiC interface, partial melting at the surface, full melting) are indicated and are clearly correlated with the characteristic shapes of the $D_{max}\left(E_{dens}\right)$ curve. We can observe first a significant increase with the fluence of the melting front nucleating at the Ni-SiC interface for $E_{dens}>E_{treshold}$, while the melting extension tends to saturate at the turning point ($E_{dens}=2.8\;{J/\mathrm{cm}}^{2}$) for the change of location of the liquid nucleation. As already noted, the melt depth increases strongly in the 2.8–3.0 J/cm^2^ interval, extending in the whole Ni-rich layer for $E_{dens}=3.0\;{J/\mathrm{cm}}^{2}$. The latest regime (i.e., the full melting one) is again characterized by an abrupt change of the $D_{max}\left(E_{dens}\right)$ curve slope, which now grows slowly with $E_{dens}$. We can understand this behavior in the full melting regime, where the possibility of additional melting is hindered by the SiC material presence (pure SiC is practically a non-melting region in our simulations). Indeed, the increased melting extension is not driven by thermodynamic energy balance only (i.e., latent heat consumption) but also by the slower interface mixing with the Ni penetrations, which lowers the local melting point.

Of course, in the full melting case, all the silicide compounds forming in the t < t_on-melt_ interval are completely dissolved since the melting phenomenon affects the whole Ni-rich layer. Similar to the partial melting case, Ni silicides form after the re-solidification of the Ni-rich layer from a local stoichiometry condition determined by the liquid phase intermixing.

In Figure 8, we show a simulation analysis of local silicide density profiles during the melting and regrowth stages of the full melting process at the 3.6 J/cm^2^ energy density; whilst the final profiles after the full quenching (at the end of the process) are shown in Figure 9. The phase function is also plotted in these figures. In this case, the Ni_3_Si, N_5_Si_2_ classes and Ni_2_Si compounds have similar weights in the silicide layer, which extends for 49 nm from the SiC interface. Again, we can observe a preferential location for the tree compounds along the depth.

In the full-melting regime, a gradual trend of silicide formation is evident from the simulated laser processes. This trend is characterized by the monotonic increase with $E_{dens}$ of the Ni-rich layer M_Ni-rich_ and the Ni-silicide M_Silicide_ region reported in Figure 10 as green and purple lines, respectively. We note that M_Ni-rich_($E_{dens}$) when $E_{dens}\geq 3.0\;{J/\mathrm{cm}}^{2}$ follows the dependence of melt depth of Figure 6 (the thickness of this layer is few nm greater than $D_{max}\left(E_{dens}\right)$ due to the already discussed Ni-penetration in the solid SiC region). In the full meting regime, we also observe a gradual increase of the weight of the Si richer Ni-silicides in mixtures residing in the M_Silicide_ region. The behaviour of M_Ni-rich_($E_{dens}$) and M_Silicide_($E_{dens}$) is not monotonic when $E_{dens}<3.0\;{J/\mathrm{cm}}^{2}$ as a consequence of the complex evolution discussed in the previous subsections. An experimental measurement of M_Ni-rich_($E_{dens}$) and M_Silicide_($E_{dens}$) for $E_{dens}=2.4,\;3.2,\;3.8\;{J/\mathrm{cm}}^{2}$ has been determined by transmission electron microscopy (TEM) for experimental laser annealing processes performed in the same conditions as the ones studied here (see Ref. [11]) and reported in Figure 10 as black crosses. Simulations and experimental analyses show a noteworthy agreement.

The excess C atoms after the silicide formation tend to segregate in the non-SiC region, and the corresponding clustering is boosted by the high liquid-phase diffusion of the residual monomer. This phenomenon is particularly important in full melting processes and at high fluences, when the high absorbed laser power maintains the liquid phase for a longer time. In order to confirm this effect, we plotted the ratio between the total C-cluster density as a function of $E_{dens}$ and the one obtained for the higher $E_{dens}=4.0\;{J/\mathrm{cm}}^{2}$ value (Figure 11); a strong increase with $E_{dens}$ is simulated for this quantity in the full melting regime. We notice that consistent with previous comments, the non-monotonic dependence of the carbon clusters density on $E_{dens}$ is related, as for the Ni-silicides case, to the switching of the initial melting location in the surface region where the local different element composition hinders the reaction yields.

As a final comment regarding this simulation analysis, we notice that in the real experimental sample, the mixture of Ni-silicides have a 3D distribution (e.g., nano-grains with given composition). Therefore, for the 1D case presented here, all the quantities have to be considered as depth-dependent averages of such distribution.

## 4. Conclusions

We have presented a method and discussed its application in a particular case, dealing with the simulation of the microstructural modification of Ni-Si-C structures caused by pulsed irradiation. We note again that the complexity of the simulation is related to the co-presence in the simulated region of different phases: elements, alloys, compounds and mixtures.

The Ni-Si-C model is integrated into the phase-field theory allowing the accurate phase transition simulation for the prediction of eventual transient kinetics of the liquid-solid regions. A preliminary parameter calibration was also proposed, and the first comparisons with experimental characterizations of the post-processed samples indicate a satisfying validity of the coupled model and parameter settings.

Numerical solutions as a function of the laser energy density are here discussed in a blanket configuration, which can also be experimentally fabricated and studied–an initial state reproducing a deposited Ni film on 4H-SiC. We have discussed and categorized the key features of the material modification promoted by the ultra-fast heating/cooling cycles, where the interplay between different phenomena (solid-phase and liquid-phase intermixing, localized melting in the Ni-SiC interface or at the surface, silicide reactions in the solid region, silicide dissolution in the liquid phase, C-clustering) play a crucial role in the stabilization of the post-process state after the total quenching of the samples.

As a result of this complex evolution, non-linear trends are also simulated for global quantities due to the typical conditions which are characteristic of different regimes: i.e., non-melting, partial-melting and full-melting process conditions. This scenario is extremely rich, and the laser parameters can be properly tuned with the aid of our simulation results to obtain different composition and space distributions of the different phases, compounds and mixtures in view of technological applications. We notice that the eventual modifications of laser parameters (e.g., shorter pulses to increase further non-equilibrium conditions) could result in the formation of new metastable phases (e.g., Ni-C carbides, Ni-Si carbides or local inclusion of other pure elemental phases in addition to the C clusters) not evident in previous experiments [10,11] with the laser pulse shown in Figure 1. In this case, the modular implementation of the model (Equations (1)–(18)) allows a fast implementation of the additional reactions and related equations necessary for simulation. The kinetic evolution of this hypothetic composite system has an enlarged number of components.

## Figures and Tables

**Figure 1 materials-14-04769-f001:**
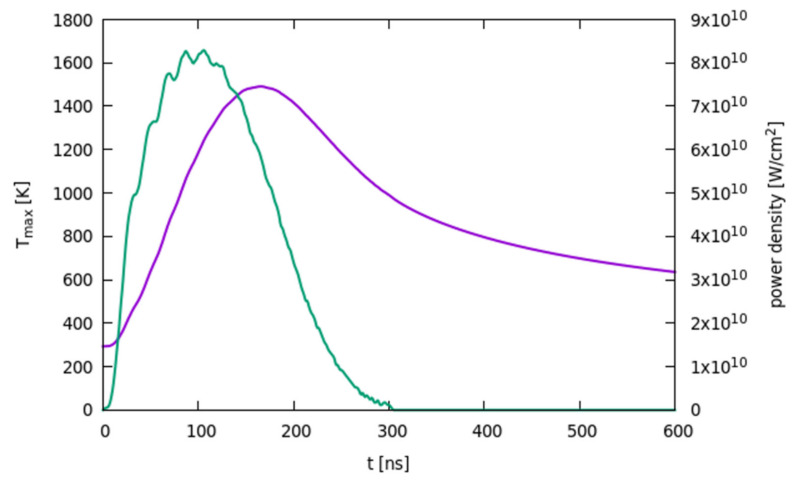
Global maximum temperature (purple line) as a function of the time obtained for $E_{\mathrm{dens}}=2.2\;J/{\mathrm{cm}}^{2}$ energy density laser process of a Ni (100 nm) + 4H-SiC stack. The power density released by the laser pulse in the Ni layer is shown as green line.

**Figure 2 materials-14-04769-f002:**
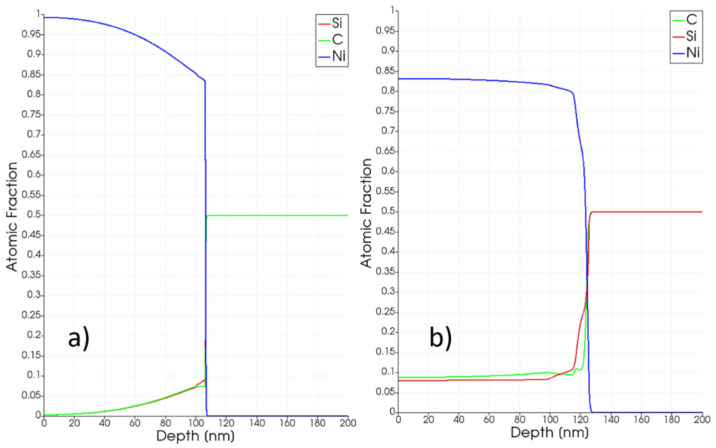
Total atomic fraction as a function of the position of the three elements of the Ni-Si-C ternary system (Ni blue line, Si red line, C green line) after 100 ns (panel (**a**)) and at the end (panel (**b**)) of the simulated irradiation at $E_{dens}=2.2\;{J/\mathrm{cm}}^{2}$ energy density.

**Figure 3 materials-14-04769-f003:**
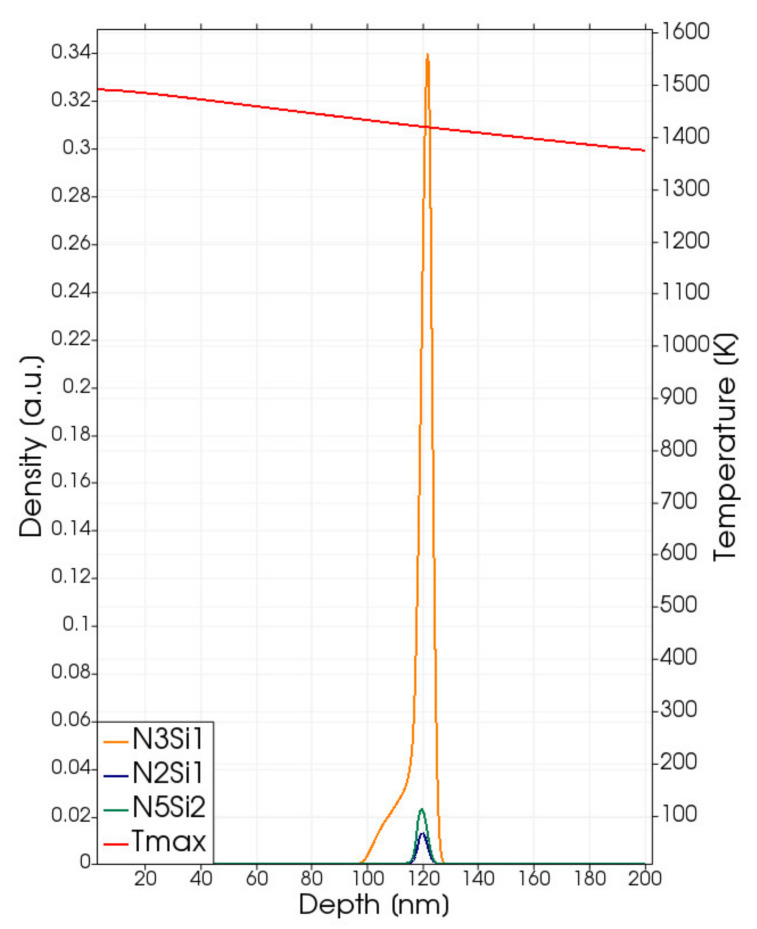
Simulated local density (left axis scale) at the end of the process of the Ni_3_Si (dark yellow lines), Ni_5_Si_2_ class (dark green lines) and Ni_2_Si (blue lines) for Laser Annealing processes with fluences $E_{dens}=2.2\;{J/\mathrm{cm}}^{2}$. Simulated local maximum temperature $T^{max}\left(x\right)$ (right axis scale), achieved in the different positions of the irradiated structure.

**Figure 4 materials-14-04769-f004:**
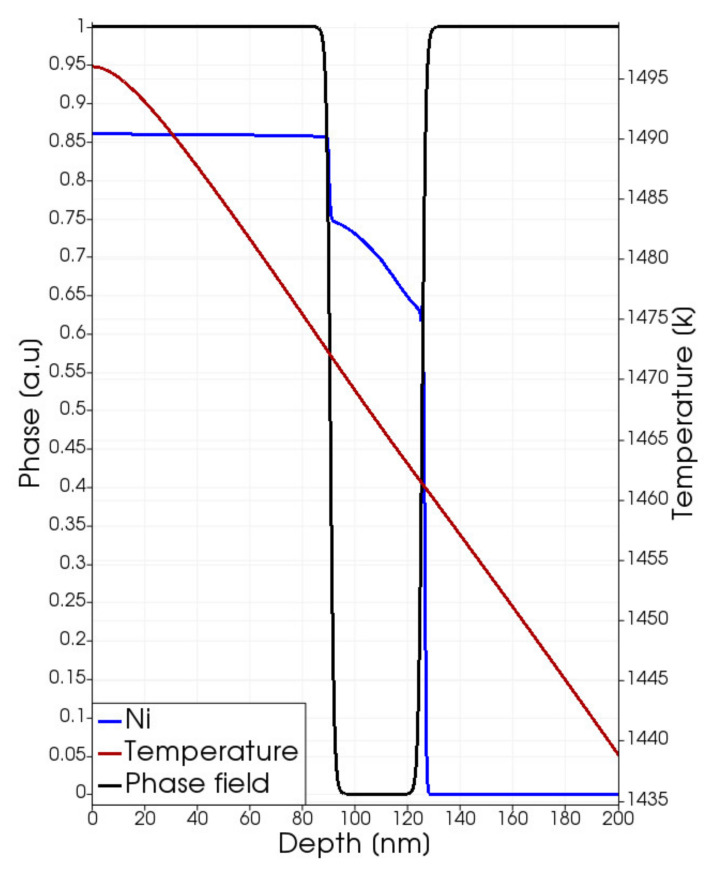
Phase-field (black line), temperature (dark red line) and Ni total atomic fraction (blue line) as a function of the position after 220 ns of the simulated irradiation at $E_{dens}=2.5\;{J/\mathrm{cm}}^{2}$ energy density. We note the phase and atomic fraction have the same range of variation.

**Figure 5 materials-14-04769-f005:**
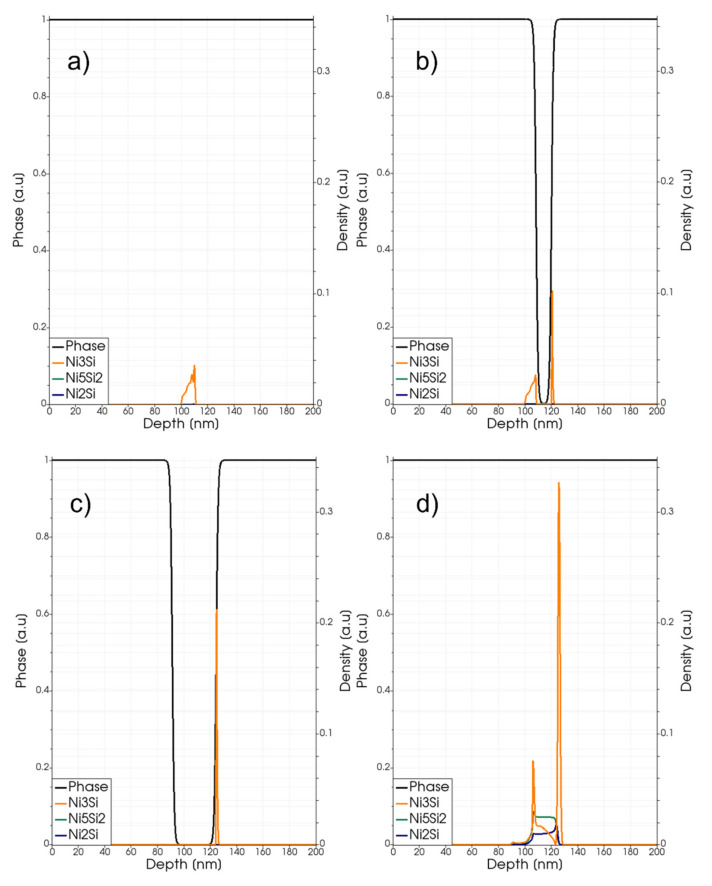
Phase-field (black line and left axis) and simulated local density (right axis scale) of the Ni_3_Si (dark yellow lines), Ni_5_Si_2_ class (dark green lines) and Ni_2_Si (blue lines) for a laser annealing process with fluence $E_{dens}=2.5\;{J/\mathrm{cm}}^{2}$. Snapshots (**a**–**d**) are taken at *t* = 120, 160, 200, 350 ns.

**Figure 6 materials-14-04769-f006:**
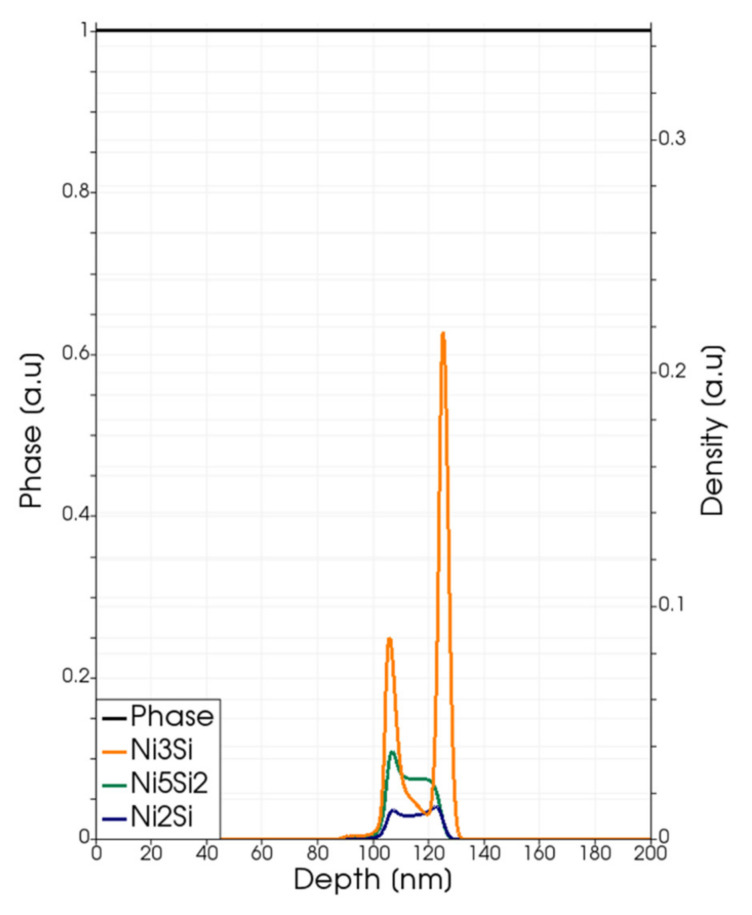
Phase-field (black line and left axis) and simulated local density (right axis scale) of the Ni_3_Si (dark yellow lines), Ni_5_Si_2_ class (dark green lines) and Ni_2_Si (blue lines) at the end of a laser annealing process with fluence $E_{dens}=2.5\;{J/\mathrm{cm}}^{2}$.

**Figure 7 materials-14-04769-f007:**
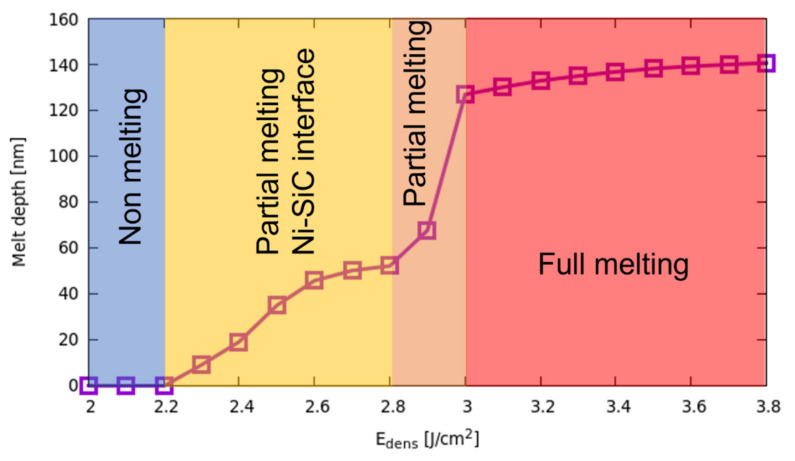
Maximum melting extension $D_{max}\left(E_{dens}\right)$ (melt depth for fluence $E_{dens}>2.8\;{J/\mathrm{cm}}^{2}$) as a function of the fluence for an irradiated Ni-4HSiC stack. The different regimes (non-melting, partial melting at the Ni-SiC interface, partial melting at the surface, full melting) are indicated by means of the colored areas.

**Figure 8 materials-14-04769-f008:**
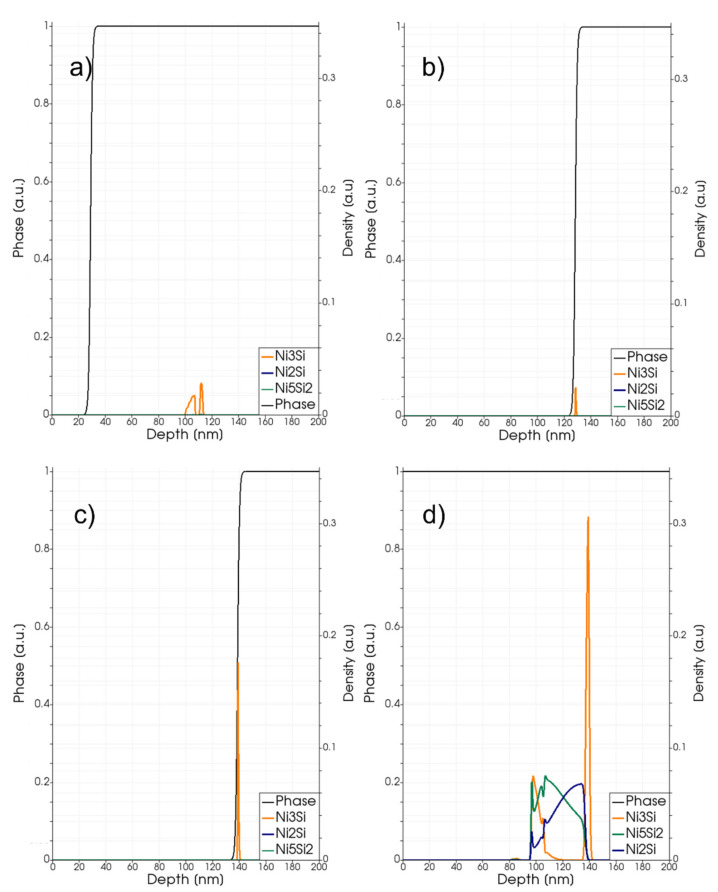
Phase-field (black line and left axis) and simulated local density (right axis scale) of the Ni_3_Si (dark yellow lines), Ni_5_Si_2_ class (dark green lines) and Ni_2_Si (blue lines) for a laser annealing process with fluence $E_{dens}=3.2\;J/{\mathrm{cm}}^{2}$. Snapshots (**a**–**d**) are taken at *t* = 120, 160, 200, 350 ns.

**Figure 9 materials-14-04769-f009:**
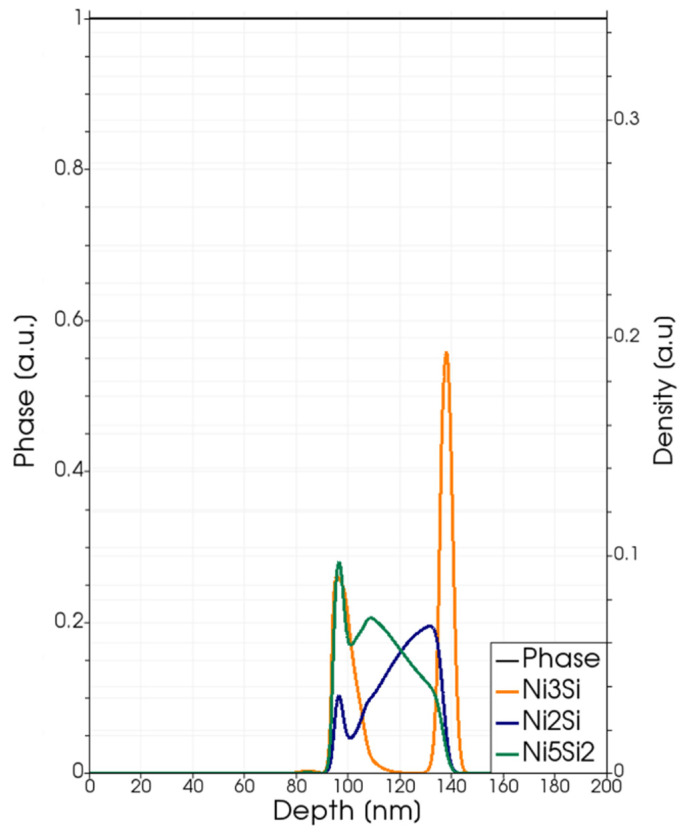
Phase-field (black line and left axis) and simulated local density (right axis scale) of the Ni_3_Si (dark yellow lines), Ni_5_Si_2_ class (dark green lines) and Ni_2_Si (blue lines) at the end of a laser annealing process with fluence $E_{dens}=3.2\;J/cm^{2}$.

**Figure 10 materials-14-04769-f010:**
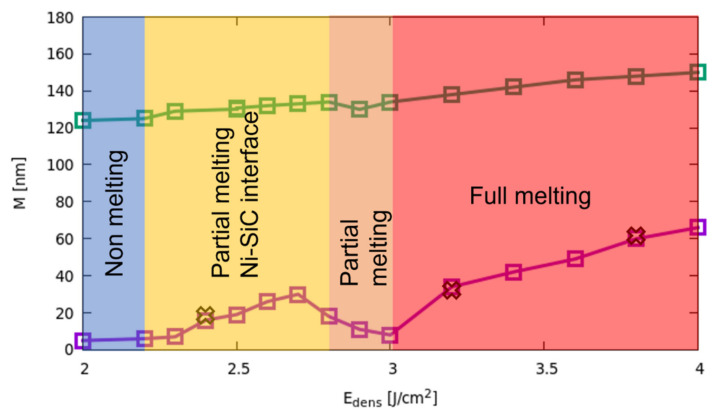
Extension of the Ni-rich layer M_Ni-rich_($E_{dens}$) (green line and square) and the extension of the silicide layer M_Silicide_($E_{dens}$) (purple line and circles) as a function of $E_{dens}$. Experimental values from [9], of the silicide layers 18, 36 and 62 nm are also reported as black crosses for the 2.4, 3.2 and 3.8 J/cm^2^ cases, respectively.

**Figure 11 materials-14-04769-f011:**
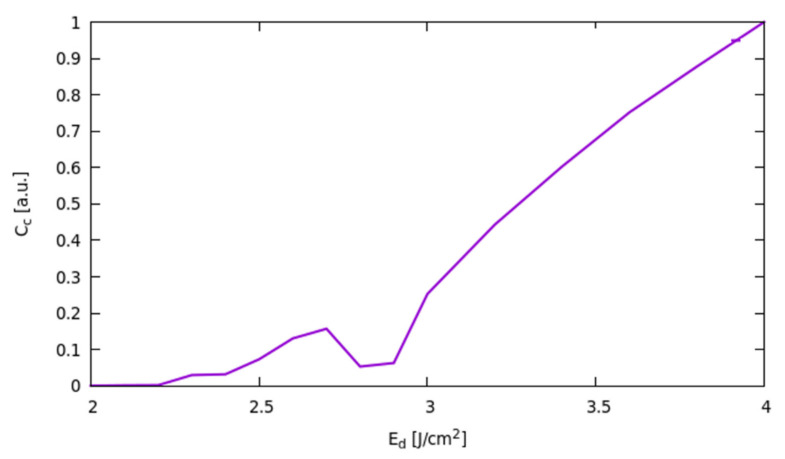
Carbon cluster density ratio as a function of the fluence of the $E_{dens}$.

**Table 1 materials-14-04769-t001:** Ni crystal material calibration parameters.

Sym. [units]	Description	Expression	Ref.
$\rho {\;[\mathrm{kg}/m}^{3}]$	Density	8902	[16]
$T_{M}\;[K]$	Melting Temperature	1728	[16]
C [J/kg K]	Thermal Capacitance	488.98 + 5.93 × 10^−3^ × T + 5.4 × 10^−5^ × T^2^	[16]
$k\;\left[W/\mathrm{mK}\right]$	Thermal Conductivity	84 + 2.13 × 10^−6^ × T/(−0.0121 + 2.29 × 10^−4^ × T + 6.1 × 10^−10^ × T^3^)	[17]
$L\;\left[{J/m}^{3}\right]$	Latent Heat	291,346	[16]
$\varepsilon_{r}\;\left(308\;\mathrm{nm}\right)$	Permittivity Real	−0.70023	[18]
$\varepsilon_{i}\;\left(308\;\mathrm{nm}\right)$	Permittivity Imaginary	8.7767	[18]
$A\;\left[m/s\right]$	Speed Pre-factor	1000	This work
$E_{a}\;\left[\mathrm{eV}\right]$	Activation Energy	−0.42	This work

**Table 2 materials-14-04769-t002:** Ni liquid material calibration parameters.

Sym. [units]	Description	Expression	Ref.
$\rho \;\left[{\mathrm{kg}/m}^{3}\right]$	Density	7902	[19]
$T_{M}\;\left[K\right]$	Melting Temperature		
$C\;\left[J/\mathrm{kg}\;K\right]$	Thermal Capacitance	735	[19]
$k\;\left[W/\mathrm{mK}\right]$	Thermal Conductivity	69	[20]
$L\;\left[{J/m}^{3}\right]$	Latent Heat	-	
$\varepsilon_{r}\;\left(308\;\mathrm{nm}\right)$	Permittivity Real	−2.66	[21]
$\varepsilon_{i}\;\left(308\;\mathrm{nm}\right)$	Permittivity Imaginary	13.17	[21]
$A\;\left[m/s\right]$	Speed Pre-factor	-	
$E_{a}\;\left[\mathrm{eV}\right]$	Activation Energy	-	

**Table 3 materials-14-04769-t003:** 4H-SiC crystal material calibration parameters.

Sym. [units]	Description	Expression	Ref.
$\rho \;\left[{\mathrm{kg}/m}^{3}\right]$	Density	3160	[22]
$T_{M}\;\left[K\right]$	Melting Temperature	3100	[22]
$C\;\left[J/\mathrm{kg}\;K\right]$	Thermal Capacitance	160 × (T ≤ 165) + (1600 – 247.16/(T − 28.38)) × (T > 165)	[22]
$k\;\left[W/\mathrm{mK}\right]$	Thermal Conductivity	[160 × 3160 × (1.895 × 10^−5^ + 8.07 × 10^−4^ × e^(−T/144)]^ × (T ≤ 165) + 3160 × [(1600 – 247.16/(T − 28.38)] × [1.895 × 10^−5^ + 8.07 × 10^−4^ × e^(−T/144)^] × (T > 165)	[22,23]
$L\;\left[{J/m}^{3}\right]$	Latent Heat	360,000	[22]
$\varepsilon_{r}\;\left(308\;\mathrm{nm}\right)$	Permittivity Real	7.2704	[22,24,25]
$\varepsilon_{i}\;\left(308\;\mathrm{nm}\right)$	Permittivity Imaginary	0.756	[22,24,25]
$A\;\left[m/s\right]$	Speed Pre-factor	1000	This work
$E_{a}\;\left[\mathrm{eV}\right]$	Activation Energy	0.42	This work

**Table 4 materials-14-04769-t004:** (Calibration obtained in the present work).

Reaction Constants	
$\frac{1}{\tau_{Ni_{3}Si}}\;[s^{-1}$]	$5\times 10^{8}\times \exp\left(-\frac{2500}{T}\right)\times \left(\frac{C^{tot}{}_{Si}}{C^{tot}{}_{Ni}}>0.2\right)$
$\frac{1}{\tau_{Ni_{5}Si_{2}}}\;[s^{-1}$]	$1.5\times 10^{8}\times \exp\left(-\frac{2900}{T}\right)\times {\left[1-{\left(\frac{C^{tot}{}_{Si}}{C^{tot}{}_{Ni}}-0.25\right)}^{2}\right]}^{-1}$
$\frac{1}{\tau_{Ni_{2}Si}}\;[s^{-1}$]	$2.0\times 10^{8}\times \exp\left(-\frac{3400}{T}\right)\times {\left[1-{\left(\frac{C^{tot}{}_{Si}}{C^{tot}{}_{Ni}}-0.29\right)}^{2}\right]}^{-1}$
$\frac{1}{\tau_{Ni_{3}Si_{2}}}\;[s^{-1}$]	$1.0\times 10^{7}\times \exp\left(-\frac{3500}{T}\right)$
$\frac{1}{\tau_{NiSi}}\;[s^{-1}$]	$1.0\times 10^{7}\times \exp\left(-\frac{3600}{T}\right)$
$\frac{1}{\tau_{NiSi_{2}}}\;[s^{-1}$]	$1.0\times 10^{7}\times \exp\left(-\frac{3800}{T}\right)$
$\frac{1}{\tau_{C-clust}}\;[s^{-1}$]	$1.0\times 10^{9}\times \left(1-\varphi +\frac{D_{X}\left(\varphi =1\right)}{D_{X}\left(\varphi =0\right)}\varphi \right)\exp\left(-\frac{3900}{T}\right)$
Intermixing coefficient *X* = *Si, Ni or C*	
$D_{X}\left[m^{2\;}s^{-1}\right]$	$2.0\times 10^{-8}\times \varphi +[8.13\times 10^{-8}\times \exp\left(-\frac{3392}{T}\right)\times \left(X\geq 0.8\right)+\;1\times 10^{-11}\times \left(0.2>X>0.8\right)+1\times 10^{-11}\times \;\exp\left(-\frac{5456}{T}\right)\times \left(X\leq 0.2\right)]\times \left(1-\varphi \right)$

## Data Availability

The data presented in this study are available on request from the corresponding author.

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
