# Peer review of "Simulations of the Ultra-Fast Kinetics in Ni-Si-C Ternary Systems under Laser Irradiation"

_materials, 2021, doi:10.3390/ma14164769_

Round 1

Reviewer 1 Report

The manuscript entitled: „Simulations of the ultra-fast kinetics in Ni-Si-C ternary systems 2 under laser irradiation” by Salvatore Sanzaro and co-workers is an example of  a well-executed theoretical study which aims at providing a comprehensive model of kinetic evolution in the sub micro-s timescale for composite materials containing regions occupied by alloys, compounds and mixtures belonging to the Ni-Si-C ternary system, which are subjected to about 100 ns long pulsed laser irradiation. The subject is timely and important as this technique is of very strong relevance to manufacturing of electronic devices and other high-tech applications. It is also vital in healing the crystal damage and in activation the impurity profile(s).

The manuscript is well prepared, clearly written and deserve to be published in Materials, after attending by the authors some of rather minor issues.

1). I’d suggest to note in the introduction that LA method is also used in research “department” to obtain materials/compounds which are otherwise hard or even impossible to form using different forms of epitaxial deposition. Doping of semiconductors with transition metals is such an important field of activities, as non-thermal-equilibrium approaches (say MBE or MOCVD) deliver only a very limited concentration range. I believe the introduction to this paper would benefit from an acknowledgement  of this another important aspect of the activity. I’d suggest to mention the following two papers: (i) Ye Yuan, R. Hübner, Fang Liu, M. Sawicki, O. Gordan, G. Salvan, D.R.T. Zahn,  D. Banerjee, C. Baehtz, M. Helm, and Shengqiang Zhou, Ferromagnetic Mn-Implanted GaP: Microstructures vs Magnetic Properties, ACS Appl. Mater. Interfaces 8, 3912−3918 (2016) – this one very nicely shows how the actual parameters of LA (say its thermal budget) influences the magnetism of the formed ternary compound; and (ii) Ye Yuan, R. Hübner, M. Birowska, Chi Xu, Mao Wang, S. Prucnal, R. Jakiela, K. Potzger, R. Böttger, S. Facsko, J. A. Majewski, M. Helm, M. Sawicki, Shengqiang Zhou, and T. Dietl, Nematicity of correlated systems driven by anisotropic chemical phase separation, Phys. Rev. Mater. 2, 114601 (2018) – this one is a very intriguing example of a completely new phase formed upon LA of InAs+Fe, i.e. the spinodal phase separation at the growth surface (that has a lower symmetry than the bulk) can lead to a quenched nematic order of this alloy components.

2) Please, explain the origin of the factor “30” in exp. 2. (line 121)

3) It is not OK write physical units in italics. So all “K”, “ns” “J”, “cm”, etc. (i) should not be slanted and (ii) there should be a single space separating the numerical value and the physical unit. E.g. instead of “1400K” it should be 1400 K (l. 256).

4) in the same line it should be … 1400 K as discussed …

5) There is something missing in line 268.

6) l. 287. It should be … calibration proposed here …

7) Figures 5 and 7 have to re-done. The legends on panels a, b, c, and d are not readable. Either these panels are removed to supplementary file and presented in a decent size or the whole concept is re-thought. For sure no one need 20 ticks around Y axes. Only 5-7 would do and the labels could be enlarged. Also the letters are 2-3 times too small. Have to at least as large as in central panel c).

8) Neither the origin of numbers given in lines 314 and 315 is known nor I do not think that magnitudes had been indeed established with below 1% (the third digits) accuracy. Does the physics of the problem is so well established that, for examples, it takes place after 121 ns? Does 1 ns “left or right” change anything? Please consider.

9) A blue line and triangles are missing in Figure 8.

10) l. 435. Should be: “… case presented here …”

Reviewer 2 Report

This work reported the simulation of annealing kinetics for Ni-Si-C system. The topic of the work was significant with regard to the development of materials manufacturing technology. The manuscript can be considered for publication after the following comments are addressed.

Comments:
(1) The existence of Ni clusters, Si clusters, Ni-C carbides, Ni-Si carbides and their possible influence on kinetics evolution should be evaluated.
(2) In Fig 7, the size of the panels abcd was too small to recognize the feature of the plots. 
